# Recent Developments in Monoclonal-Antibody-Based Biologic Therapy for Severe Refractory Eosinophilic Asthma

**DOI:** 10.3390/antib14040101

**Published:** 2025-11-25

**Authors:** Garry M. Walsh

**Affiliations:** Institute of Medical Sciences, University of Aberdeen, Foresterhill, Aberdeen AB25 2ZD, UK; g.m.walsh@abdn.ac.uk; Tel.: +44-(0)-7747-761204

**Keywords:** asthma, IL-4, IL-13, IL-5, eosinophils, biologics

## Abstract

Background: Asthma exhibits marked heterogeneity both clinically and at the molecular phenotypic level, requiring specifically targeted treatments to block the key pathways of the disease. Monoclonal-antibody-based biologics targeted at critical inflammatory pathways of T2 inflammation such as IL-5, IL-5R, IL-4, and IL-13 are increasingly regarded as effective treatments for severe refractory eosinophilic asthma. Methods: This review provides an update on the potential of straightforward and reproducible biomarkers to aid in the selection of the biologic-based therapy most likely to be effective in patients with severe or refractory eosinophilic asthma based on English-language original articles in PubMed or MedLine. Results: Monoclonal-antibody-based biologic therapies have revolutionised severe asthma management, enabling reductions in symptoms that include exacerbations, discontinuation of oral corticosteroids, improved lung function, and enhanced quality of life. Significant clinical effects with anti-IL-5 or -IL-4/13 monoclonal antibodies are more likely to be seen when simple predictive biomarkers such as serum periostin, fractional exhaled nitric oxide (FENO), or blood eosinophil counts are used to aid in the identification of those patients with severe refractory eosinophilic asthma who are most likely to benefit from biologic therapies. Conclusions: Biologic-based therapy aimed at T2 inflammation benefits patients with severe eosinophilic asthma, particularly when guided by biomarkers that do not require direct sampling of the airways to target therapy, who are most likely to benefit from these treatments, with good safety profiles for these therapies.

## 1. Introduction

Asthma is a chronic inflammatory airway disorder belonging to obstructive respiratory diseases and is characterised by variable airflow limitation, with clinical severity ranging from very mild and occasional symptoms to recurrent, significant exacerbations that, in a small number of patients, can result in a near-fatal or fatal outcome. Significant symptoms include chest tightness, cough, and wheeze, together with airway hyperresponsiveness and airway remodelling [1]. For most treatment-compliant patients, close-to-complete disease control or even remission is achievable through conventional, well-established step-based pharmacotherapy, with inhaled corticosteroids (ICSs) providing the key foundation for asthma treatment [2]. However, approximately 5% of people with asthma are considered to have severe or refractory treatment-resistant disease when high-intensity treatment is required to maintain control or symptoms remain uncontrolled despite such intervention, representing a clear unmet medical need with significant health cost implications [3,4]. If symptom control in severe asthma is not achieved with standard of care (high doses of ICSs/long-acting β2-adrenergic agonist (LABA) combinations), together with the addition of other drugs such as leukotriene modifiers, guidelines recommend treatment escalation to maintenance oral corticosteroids (OCSs) [5]. However, an important concern is that increasing the use of OCSs is associated with a wide range of well-recognised and serious short- and long-term adverse events [6], resulting in significant morbidity and quality of life issues [7]. Importantly, OCSs fail to fully prevent the IL-33-mediated airway structural changes associated with airway remodelling [8].

Many decades of research into the processes that underlie asthma have revealed different patterns of cytokine-based inflammation involving diverse cell types such as T cells, B cells, mast cells, eosinophils, basophils, neutrophils, and dendritic cells, as well as structural cells including epithelial, smooth muscle, and mesenchymal cells. The increased understanding of such complex inflammatory pathways has resulted in the development of antibody-based biological therapies that target T2 cytokines such as IL-4, IL-5, and IL-13, which play key roles in the inflammatory processes underlying asthma [9]. Importantly, straightforward discriminatory non-invasive biomarkers aid in asthma phenotype identification, which in turn guides the choice of the appropriate biologic to ensure meaningful treatment outcomes for patients [10,11]. This review summarises recent original reports in PubMed or Medline on the utility of simple reproducible biomarkers to aid in the selection of biologics specific for IL-4, IL-5, and IL-13 in the treatment of severe refractory eosinophilic asthma.

## 2. Asthma Phenotypes

Asthma can be considered in terms of ‘asthma phenotypes’, which take demographic, clinical, and/or pathophysiological characteristics into account [12], including allergic asthma, non-allergic asthma, late-onset adult asthma, asthma with persistent air flow limitation and obesity-associated asthma [13]. Disease entities termed ‘asthma endotypes’ provide a further refinement based on specific pathophysiological mechanisms. The two primary endotypes that take pathophysiological mechanisms into account at the cellular and molecular levels are type 2 (T2) and non-type 2 (non-T2) asthma. T2-high asthma is seen in around 50% of patients, who typically have eosinophilic inflammation mediated by cytokines including IL-4, IL-5, and IL-13, with enhanced levels of immunoglobulin E (IgE) often present. In contrast, the non-T2 endotype is seen in non-eosinophilic asthma and encompasses a more complex group of patients without T2 characteristics, with a significant neutrophilic inflammation often observed in patients who exhibit CS resistance with frequent severe or refractory exacerbations [14,15]. Acute exacerbations in non-T2 asthma often mimic those of T2 asthma in terms of clinical presentation and symptoms but are marked by a greater heterogeneity in inflammatory mediators, involving inflammatory cells such as Th17 cells, innate lymphoid cell types (LCs), and M1 macrophages [16,17]. The picture is frequently further complicated by the presence of overlapping disease coexisting with asthma in some patients such as chronic obstructive pulmonary disease (COPD), rhinosinusitis, or gastroesophageal reflux disease (GERD), thereby further exacerbating symptoms [18,19,20].

## 3. Biomarkers

Our understanding of the complex patterns of inflammation underlying asthma pathogenesis has informed the development of antibody-based biological therapies that target the T2 cytokines, IL-4, IL-5, and IL-13, in adults [21,22,23] and in adolescents and children with refractory asthma [24]. Meaningful clinical effects with anti-cytokine-based biologic therapies require carefully selected patient populations that take asthma phenotypes into account, guided by straightforward non-invasive discriminatory biomarkers, such as blood eosinophil counts, FeNO, and serum periostin [25,26]. To be clinically useful biomarkers must aid in disease monitoring, diagnosing, predicting treatment responses, and tracking the advancement of the disease [27,28].

Peripheral blood and sputum eosinophil counts correlate with disease activity in both allergic and non-allergic asthma, where their increased numbers signal greater severity. The peripheral blood eosinophil count is considered to be one of the most reliable, straightforward and methodologically simple biomarkers for classifying asthma subtypes, with serial blood measurements routinely used for monitoring the response to anti-IL-5 therapies [29]. Airway epithelium synthesis of nitric oxide is upregulated by IL-4 and IL-13 signalling through the IL-4Rα receptor. FeNO measurement represents a non-invasive readout of eosinophilic inflammation and type-2-driven asthma that can be carried out in-home to aid therapy adjustments, which, together with blood eosinophil levels, is the cornerstone of biomarker-guided management in severe asthma [30]. Furthermore, serum periostin is an extracellular matrix protein released by airway epithelial cells stimulated with IL-13 that has potential as a biomarker of airway eosinophilia in patients with asthma [31].

## 4. IL-5

The role played by eosinophils in the pathogenesis of asthma is well-recognised [32], with their substantial arsenal of granule-derived basic proteins, lipid mediators, cytokines, and chemokines significantly contributing to airway inflammation and lung tissue remodelling, characterised by airway thickening, fibrosis, and angiogenesis [33,34], leading to increased disease severity, exacerbation frequency, and decreased lung function [35,36]. Importantly, repeated exacerbations in patients with severe eosinophilic asthma may be associated with accelerated loss of lung function [37]. IL-5 plays an essential role in the maturation and release of eosinophils from the bone marrow, together with the tissue accumulation, activation, and persistence of eosinophils through the inhibition of apoptosis. This understanding of the effects of IL-5 on eosinophil function led to the development of the humanised anti-IL-5 mAb mepolizumab and reslizumab [38,39], while benralizumab is a mAb specific for the a chain of the human IL5 receptor [40,41].

### 4.1. Mepolizumab

The anti-IL-5 IgG1 mAb mepolizumab (“Nucala”, GlaxoSmithKline) is approved for the treatment of severe eosinophilic asthma at a recommended dose of 100 mg every 4 weeks [42,43] and in other eosinophilic disorders including chronic rhinosinusitis with nasal polyps, eosinophilic esophagitis, and hypereosinophilic syndrome [44]. Mepolizumab efficacy in the treatment of severe eosinophilic asthma is supported by a considerable body of data from randomised controlled trials, together with accumulating evidence from real-world studies that further demonstrate its efficacy and safety. Mepolizumab treatment outcomes in patients with refractory asthma include significant reductions in exacerbation rates, enhanced health-related quality of life parameters, with moderate improvements in symptom scores and forced expiratory volume in 1 s (FEV1), and a positive safety profile [45,46,47,48]. Meta-analyses of mepolizumab treatment of patients with severe eosinophilic asthma reported reduced exacerbation rates, improved quality of life [49], and fewer hospitalisations, and/or emergency care due to exacerbations [50], positive clinical outcomes associated with eosinophil blood counts greater than 150 cells/μL [51], demonstrating the usefulness of this simple biomarker for mepolizumab treatment. In this regard, McDowell and colleagues [16] reported that, in the MEX study of patients with severe eosinophilic asthma treated with mepolizumab, significant reductions in circulating eosinophils were reported. However, approximately half of these patients had residual attacks driven by eosinophilic inflammation, while the other half showed a non-eosinophilic profile. Importantly, FeNO measurements taken during an exacerbation distinguished high-eosinophil from low-eosinophil events, suggesting that exacerbations are two distinct entities, i.e., non-eosinophilic events are driven by infection associated with low FeNO and high C-reactive protein concentrations, whereas eosinophilic exacerbations were associated with high FeNO concentrations. The authors suggested that the results of their study question the routine use of OCSs for the treatment of all asthma exacerbations during mepolizumab treatment and that switching of biological therapies in the event of treatment failure should be preceded by profiling the inflammatory phenotype of ongoing asthma exacerbations.

The majority of clinical trials report that mepolizumab improved primary outcomes within six months of commencing treatment. For example, a prospective cohort of treatment with subcutaneous mepolizumab of 20 subjects with severe eosinophilic asthma [52] reported significant effects on lung function, quality of life, and improvement in ventilation inhomogeneity within 4 weeks and was sustained for six months. Furthermore, there is increasing evidence of a CS-sparing effect following mepolizumab treatment [53,54,55]. The primary outcome of the COSMEX trial [56] was to assess the long-term safety and efficacy of mepolizumab in patients with severe eosinophilic asthma. Although assessment of OCS use was a secondary outcome, 38 patients with more than 128 weeks of continuous use reported reducing their daily OCS use, while 45% of patients discontinued OCS use entirely. Taken together, these studies support mepolizumab as a long-term treatment resulting in a slow but progressive tapering of OCS use in patients with severe asthma. In this respect, a single case report of a patient with near-fatal severe eosinophilic asthma, who did not respond to treatment with omalizumab, described positive effects of mepolizumab treatment [57]. Moreover, a real-life retrospective cohort study of 346 patients with severe asthma found that mepolizumab significantly reduced exacerbations and OCS and ICS use [58]. A retrospective observational study over 24 months of mepolizumab treatment in patients with severe eosinophilic asthma reported reduced exacerbations, improved lung function and asthma control, and a 62% reduction in the number of patients requiring OCS, while doses were reduced (7.8 mg/day versus 20.6 mg/day) for patients who required OCS treatment [59]. Similar outcomes have been observed in several real-world studies, including OCS-sparing effects following treatment with mepolizumab [60,61,62,63,64,65,66,67]. Furthermore, a systematic literature review and indirect treatment comparison [68] reported mepolizumab is as effective as omalizumab in improving lung function in patients with severe asthma. Overall, information from clinical trials and post-market surveillance suggests a good safety profile for mepolizumab [69]. More recently a multicentre, Phase IIIb safety, open-label extension study of multiple prior studies that examined long-term mepolizumab treatment in addition to standard of care reported a favourable safety profile for mepolizumab, consistent with previous reports, with most patients having a favourable benefit:risk ratio for up to 10 years [70]. Out of 514 enrolled patients, 24 were aged between 6 and 17 years, thereby suggesting a positive risk–benefit balance for paediatric patients. In terms of efficacy and safety of mepolizumab, a recent case series study assessed 12 months of mepolizumab therapy in four paediatric patients aged 10 to 16 years with severe eosinophilic asthma. A significant reduction in blood eosinophil count was accompanied by a decrease in asthma exacerbation frequency by over 50%, with significantly improved ACT across all subjects. However, there were no significant pre- or post-treatment changes in FEV1 [71]. Another study examined the utility of mepolizumab in exacerbation-prone children and adolescents with eosinophilic asthma, who were randomly assigned to receive mepolizumab (146 patients) or placebo (144 patients) every 4 weeks for 52 weeks. Compared with placebo, mepolizumab significantly reduced the annualised rate of asthma exacerbations. However, significant differences between groups in secondary outcomes were lacking, such as time to first exacerbation, lung function, or patient-reported asthma severity scores. Mepolizumab reduced T2 and eosinophil-associated inflammatory pathways, together with eosinophil cytoplasmic proteins associated with higher risk of exacerbations in the placebo group, with upregulation of epithelial inflammatory pathways and IL-33 responses in the mepolizumab group correlated with persistent exacerbation risk. Apart from injection site reactions, mepolizumab was generally well-tolerated [72].

### 4.2. Reslizumab

Reslizumab (CinquilTM, Teva Pharmaceuticals, Petah Tikva, Israel) is an IL-5 neutralising IgG4K mAb used as an add-on maintenance therapy for adults with severe eosinophilic asthma administered intravenously once every 4 weeks at 3 mg/kg [9]. Two parallel multicentre, double-blind, Phase 3 trials that recruited OCS-treated patients with a blood eosinophil count greater than 400 cells/µL and one or more exacerbations in the previous year demonstrated improved FEV1, forced vital capacity, ACQ, and reductions in the use of rescue short-acting β2-agonists [73]. Another Phase 3 study reported that reslizumab treatment significantly increased mean FEV1 in patients with poorly controlled asthma with eosinophil counts greater than 400 cells/µL [74]. Two doses of reslizumab (0.3 and 3.0 mg/kg for 16 weeks) in 311 patients with poorly controlled asthma with elevated blood eosinophil levels improved FEV1 and ACQ score and decreased inhaled rescue medication use, together with significant increases in FEF25–75 (forced expiratory flow at 25–75% of forced vital capacity) [75]. A post hoc analysis reported significantly greater reductions in asthma exacerbations and improved lung function in patients with poorly controlled late-onset eosinophilic asthma given reslizumab every 4 weeks compared with patients with early-onset asthma. The authors suggested identification of a reslizumab-responsive late-onset asthma subgroup from the overall population but acknowledged a requirement for further prospective controlled studies to support these observations. A long-term, international, multicentre, non-randomised, open-label extension study in adolescent and adult patients with moderate-to-severe eosinophilic asthma observed that reslizumab treatment improved lung function and asthma control for up to two years. Reslizumab was well-tolerated, thereby supporting its role in long-term control of moderate-to-severe refractive eosinophilic asthma [76].

The efficacy and safety of reslizumab in severe eosinophilic asthma has also been demonstrated in real-world studies. One two-year study examined the real-world treatment of 26 patients with severe asthma who averaged of 8.3 exacerbations in the preceding year, over half of who were taking OCS, and 30% required hospitalisation. The study reported that asthma control significantly improved, with a reduction in annual exacerbation frequency of 79% in the first year and 88% in the second year, while more than a third of patients discontinued OCS treatment [77]. A retrospective study of 215 patients treated with reslizumab reported positive clinical and patient-reported outcomes including improved lung function together with reductions in OCS use over 6 months maintained over a 12-month period [78]. Another multicentre, retrospective, real-life study reported that treatment with reslizumab achieved complete asthma control in 40% of 208 patients with severe eosinophilic uncontrolled asthma, with significant reductions in exacerbations and OCS dose and meaningful improvements in symptoms, with an adequate safety profile [79]. Another retrospective study reported that reductions in severe exacerbations and OCS use were seen in biologic-naïve reslizumab patients or those switched from another type 2 biologic [80]. These authors concluded that patients who are non-responsive to one type 2 biologic should be switched to an alternative, even if it targets the same molecular pathway.

### 4.3. Benralizumab

Benralizumab (AstraZeneca) is a humanised afucosylated IgG1k isotype indicated as add-on maintenance treatment of individuals aged 12 years and above with uncontrolled severe eosinophilic asthma with an absolute eosinophil count exceeding 300 cells/μL [81]. The Phase 3 trials SIROCCO and CALIMA [82,83] reported that add-on subcutaneous benralizumab in patients with severe uncontrolled asthma who continued their normal inhaled therapy had depleted blood eosinophils, significantly reduced annual rate of severe exacerbations, and improvements in symptoms and FEV1. A further analysis of these two studies found that benralizumab reduced asthma exacerbation rates by 42% in SIROCCO and by 36% in CALIMA in patients with blood eosinophil counts greater than 150 cells/μL, together with significant improvements in pre-bronchodilator FEV1 and total asthma symptom scores [84]. An optimal benralizumab dosage of 30 mg every 8 weeks with an additional dose at week 4 was confirmed for patients with severe eosinophilic asthma by a pharmacokinetic study based on SIROCCO and CALIMA [85], while the ZONDA Phase 3 study reported that subcutaneous (30 mg) benralizumab produced significant reductions in OCS use in patients with severe eosinophilic asthma [86].

The SOLANA [87] multicentre Phase 3b trial reported that subcutaneous benralizumab produced non-significant improvements from baseline in prebronchodilator FEV1, FVC, and whole-body plethysmography in patients with severe eosinophilic exacerbation prone asthma with baseline blood eosinophil counts ≥ 300 cells/µL. Interestingly, in those patients with nasal polyps receiving benralizumab, a subgroup analysis demonstrated substantial improvements in prebronchodilator FEV1 from baseline and SGRQ scores. These observations are supported by findings from the SIROCCO and CALIMA trials, in which benralizumab produced superior improvements in lung function and ACQ-6 in patients with nasal polyps compared with those without nasal polyps [88]. The BORA Phase 3 extension study [89] enrolled adults who completed SIROCCO, CALIMA, or ZONDA to continue treatment for up to 56 weeks (adults) or for up to 108 weeks (adolescents) and found that the safety profile of benralizumab was in agreement with published results from the Phase 3 studies, findings further supported by an extension study in adolescents over 2 and 3 years of treatment [90]. Data from the BORA and ZONDA studies [91] demonstrated significant reductions in OCS usage over 18 months of benralizumab treatment in adult patients with severe asthma. Post hoc analysis of data from the SIROCCO and CALIMA studies of patients with fixed airway obstruction associated with uncontrolled eosinophilic asthma reported improvements with benralizumab in asthma exacerbation rates, lung function, asthma symptoms, and health-related quality of life [92]. Further post hoc analysis of pooled data from SIROCCO and CALIMA demonstrated improved post-bronchodilator lung function in patients with eosinophilic asthma and that elevated blood eosinophil counts, presence of nasal polyposis, decreased lung function, and long-term OCS use were all potential indicators of improved responses to benralizumab treatment, with a baseline FVC of less than 65% predicted as one of the three most significant clinical characteristics indicative of an enhanced response to benralizumab. These authors opined that benralizumab treatment might reverse some structural changes in the airways that are associated with eosinophil-driven chronic inflammation that may not be always detectable by blood eosinophil counts alone [93].

A cross-sectional, multicentre, real-life study of 42 consecutive patients with severe refractory eosinophilic asthma reported improvements in asthma control, lung function, together with reductions in ICS and OCS use and fewer emergency department visits following benralizumab treatment [94]. A similar cohort observational study of 18 patients with severe eosinophilic asthma reported reduced exacerbations, hospitalisations, and significant improvements in ACQ, together with a CS-sparing effect after 26 weeks of treatment. At 52 weeks post-treatment, there was a 26.8% increase in FEV1 from baseline, while those patients with nasal polyposis had an increase of nearly 50%, and those with blood eosinophil count greater than 500 cells/μL showed an increase of 68% [95]. These real-life studies support clinical trials, demonstrating benralizumab to be an effective and safe treatment for difficult-to-treat eosinophilic asthma. Overall, treatment with anti-IL-5 biologics is effective in patients with eosinophilic asthma through exacerbation prevention, with accumulating evidence of CS-sparing effects associated with acceptable safety profiles. These biologics are also effective in other eosinophil-related conditions such as eosinophilic esophagitis, eosinophilic granulomatosis with polyangiitis, and chronic eosinophilic pneumonia [96,97,98]. One other important issue is that while type-2-directed therapies have significant beneficial effects in some patients with severe asthma, mainly through reductions in the rate of acute exacerbations, these are not fully eliminated. Furthermore, more modest effects are evident on other clinical metrics including lung function, symptoms, and quality of life [99].

## 5. IL-4 and IL-13

In addition to their pivotal role in IgE synthesis by B cells, IL-4 and IL-13 promote the differentiation of naïve T cells into Th2 effector cells, together with playing key roles in eosinophil accumulation, including stimulation of epithelial and endothelial cells to secrete chemokines such as eotaxin-3, a potent chemoattractant for eosinophils to inflamed tissues. IL-13 also drives goblet cell hyperplasia, mucus hypersecretion, smooth muscle contraction, and hypertrophy, while IL-4 has a proven role in upregulating expressions of endothelial adhesion molecules [100,101]. Thus, both cytokines play vital roles in the T2 immune responses that are observed in asthma, allergic rhinitis, food allergies, and atopic dermatitis.

### Dupilumab

Dupilumab (SAR231893/REGN668, Regeneron Pharmaceuticals Inc., Tarrytown, NY, USA) is a VelocImmune-derived fully humanised mAb against the IL-4Ra chain, the shared receptor component for IL-4 and IL-13, administered at 200 or 300 mg. An early Phase 2A study of patients with moderate-to-severe asthma with 300 eosinophils/μL or a sputum eosinophil level greater than 3% reported that dupilumab significantly reduced exacerbations, serum IgE, eotaxin-3, TARC, and the biomarker FeNO, with significant increases from baseline in FEV1 seen at week 2, which continued through to week 12, although patients had been instructed to taper and stop inhaled glucocorticoids during weeks 6 to 9 [102]. A similar study of dupilumab as an add-on therapy in 769 patients with a blood eosinophil count greater than 300 cells/μL reported significant improvements in lung function and patient-reported asthma symptom outcomes and reduced exacerbation rates, which were evident in the overall population and in the subgroup of patients with fewer than 300 eosinophils per microlitre [103]. Quest [104], a Phase 3 study of 1902 patients aged over 12 years with uncontrolled moderate-to-severe asthma, reported that dupilumab reduced serious asthma exacerbations with the increase in FEV1 being greater in patients with a baseline blood eosinophil count > 300 cells/μL or baseline FeNO of >25 ppb. Importantly, analysis of the slope demonstrated that, compared with the placebo group, dupilumab prevented annual reduction in postbronchodilator FEV1, suggesting potential for this biologic to reduce airway remodelling. LIBERTY ASTHMA VENTURE [105], an international Phase 3 trial of patients with glucocorticoid-dependent severe asthma that accompanied Quest, found that dupilumab significantly reduced OCS use together with significant reductions in severe exacerbations and improvements in FEV1. The most robust results from both the Phase 3 studies were seen in patients with increased T2 immune activity, including elevated blood eosinophil counts and FeNO levels. Interestingly, a post hoc analysis of LIBERTY ASTHMA Quest evaluated whether baseline FeNO, adjusted for baseline blood eosinophil levels and other clinical characteristics, could act as an independent predictor of dupilumab responses to treatment in patients with uncontrolled moderate-to-severe asthma. Increased baseline FeNO levels were found to be associated with improved clinical benefits in patients receiving dupilumab compared with placebo. The authors concluded that, as patients with elevations in FeNO and/or eosinophils are particularly likely to respond to dupilumab, these should be measured prior to considering the use of a biologic for severe asthma [106].

These and other studies further confirm key roles for both IL- 4 and IL-13 in T2 inflammation and that dupilumab improved lung function, exacerbation rates, and patient-reported outcomes and reduced use of OCS in patients with uncontrolled persistent asthma [107], considerations emphasised by real-world studies of the efficacy of dupilumab [108]. Mucus plugs are seen on chest CT scans in over half of patients with severe asthma, and these are associated with airflow limitation that can persist even with inhaled or systemic steroid administration [109,110]. It is of interest therefore that a single case study of a patient with asthma and mucus plugging who was treated with dupilumab 600 mg, followed by 300 mg every 2 weeks, reported that, in addition to improvements in the asthma control test and FEV1, a subsequent chest CT scan demonstrated a reduction in the mucus plug score from 9 to 3 [111]. IL-13 is a key cytokine involved in the formation of mucus plugs [100]; there is therefore a need for large-scale studies examine whether dupilumab is an effective therapeutic option for mucus plugs in patients with asthma.

Dupilumab has been approved for the therapy of moderate-to-severe asthma in children older than six years and adolescents [112], particularly in patients with a T2 inflammatory phenotype characterised by elevated blood eosinophils or FeNO [113,114]. For example, the Voyager phase III RCT addressed the effectiveness of add-on subcutaneous dupilumab administered bimonthly for 52 weeks to 408 children (6–11 years) with uncontrolled moderate-to-severe asthma; of these, 86% exhibited a T2 inflammatory phenotype, demonstrated by an elevated eosinophil count (≥150 cells/μL) or FeNO (≥20 ppb). For primary outcomes, the authors reported significant reductions in the annualised rate of severe exacerbations or hospitalisations, with the greatest effect highest observed in children presenting the most pronounced T2 phenotype, characterised by peripheral eosinophils of ≥300 cells/μL. Secondary outcomes showed that dupilumab treatment significantly improved FEV1 values and asthma control, measured by the seven-item Asthma Control Questionnaire (ACQ-7), which were also greater in children the T2 phenotype, with a similar adverse effect profile in the two groups [115]. Dupilumab also produced rapid and sustained decreases in the markers of T2 inflammation FeNO, serum total IgE and serum thymus, and activation-regulated chemokine (TARC) [116]. Two further studies that analysed data from Voyager reported improved asthma control and health-related quality of life (HRQoL) [117] and significant improvements in FVC, FEV1/FVC, and FEF25–75 both pre-bronchodilation and post-bronchodilation [118]. These observations are supported by a recent systematic review that also concluded that dupilumab treatment is effective in reducing the annual rate of severe exacerbations, with improved quality of life and pulmonary function parameters and minimal serious adverse events in children and adolescents with moderate-to-severe asthma [114].

Multiple studies of dupilumab in other type 2 inflammatory conditions such as chronic sinusitis, nasal polyposis, eosinophilic esophagitis, and atopic dermatitis (reviewed in [119,120,121,122] demonstrate it to be a highly effective biologic-based therapy for a wide range of allergic diseases.

## 6. Conclusions

The inherent heterogeneity in the complex inflammatory processes that underly asthma pathogenesis has long been a clinical challenge, as anti-inflammatory and bronchodilator therapy regimens fail to fully attenuate these diverse immunological pathways, particularly in severe disease. The advent of biologic therapies represents a paradigm shift that enables the direct targeting of the key molecular drivers that orchestrate airway inflammation, remodelling, and disease exacerbations. Straightforward, non-invasive biomarkers such as serum periostin, FENO, and blood eosinophil counts are key elements in guiding the effective application of biologic-based therapy aimed at T2 inflammation, thereby benefiting patients, including children and adolescents, with eosinophilic asthma [123]. Anti-IL-5 biologics reduce exacerbations and have CS-sparing effects in these patients. Similar outcomes were reported for targeting of IL-4/13 by biologics such as dupilumab, where a baseline blood eosinophilia to aid in patient selection was not always a pre-requisite. However, treatments aimed at T2 inflammation provide little benefit to patients with T2-low asthma. Several novel and promising biologics are under development [124]. These include depemokimab, a next-generation anti-IL-5 mAb; itepekimab, a fully human IgG4P mAb that targets the alarmin IL-33; and stapokibart, which is a humanised mAb directed against IL-4Rα, thereby inhibiting both IL-4 and IL-13 signalling [23].

Tezepelumab, a mAb specific for thymic stromal lymphopoietin, reduced clinically significant asthma exacerbations and symptom severity in patients with severe asthma patients. Importantly, these positive outcomes were independent of the underlying asthma endotype or phenotype and were also irrespective of baseline biomarkers, such as blood eosinophil count and FeNO [125,126]. There is also interest in the IL-17 family of cytokines that can trigger neutrophilic inflammation and airway remodelling in severe asthma [127]. However, treatment with the humanised mAb brodalumab, which blocks IL-17RA, did not result in symptom improvement in patients with inadequately controlled moderate-to-severe asthma who continued their ICS [128]. IL-33 has been associated with asthma susceptibility and can contribute to both T2 and non-T2 inflammation; it is therefore of interest that treatment of patients with severe asthma with astegolimab, a human IgG2 inhibitor the IL-33 receptor ST2, reduced annualised asthma exacerbation rates compared with placebo [129]. Furthermore, a placebo-controlled four-arm trial reported that treatment with itepekimab improved asthma control and lung function [130]. These studies suggest that targeting the IL33 pathway is a promising therapeutic option.

A major consideration is that the cost of biologic-based therapy can be significant [130]. In this respect, the estimated cost effectiveness of mepolizumab use exceeds value thresholds, which may need significant reductions in the current list price [131]. Furthermore, while a systematic review reported the effectiveness of benralizumab, dupilumab, mepolizumab, omalizumab, and reslizumab in the treatment of eosinophilic asthma and that positive treatment outcomes could result in savings due to decreased hospitalisations or primary care visits, the incremental cost-effectiveness ratio per quality-adjusted life year value is above the willingness-to-pay threshold [132]. These considerations emphasise the importance of encouraging patient compliance with current CS-based therapy to control symptoms before moving unnecessarily to expensive biologic-based therapies.

## 7. Future Directions

For future research there is a requirement for more clinical trials in children and adolescents, together with head-to-head comparisons of biologic efficacy, with an overall aim of achieving durable disease remission in all patients. The potential for the identification of novel predictive biomarkers does appear rather limited in terms of improving treatment outcomes with the currently available biologics that target T2 pathways. In the longer term, the integration of emerging novel biologic agents that result in significant symptom relief without the requirement for measurement of baseline biomarkers as part of treatment algorithms has significant potential to deliver personalised, mechanism-based care for patients with severe refractory asthma [133].

## Data Availability

No new data were created or analyzed in this study.

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
