# Peer review of "Recent Developments in Monoclonal-Antibody-Based Biologic Therapy for Severe Refractory Eosinophilic Asthma"

_2073-4468, 2025, doi:10.3390/antib14040101_

Round 1

Reviewer 1 Report

Comments and Suggestions for Authors

This review is of clinical interest. The following issues should be considered:

  1. Abstract can be shortened to an essential preview.
  2. A graphical abstract of Mab biological therapies in severe asthma may allow a quick grasp of the use of Biologics.
  3. Please add a table of the major mab based biologics.
  4. Glossary of the different, therapeutic antibodies with the target cytokine or receptors is desirable, ev as a separate box.
  5. Are there any biologic Mabs to neutralise chemokines, another potential target for severe asthma? What about IL-33 or its receptor ST2?
  6. Severe asthma: What are the key cytokine/chemokines?
  7. Role of IL-17 family members, involved in severe asthma? Should bee discussed.
  8. Add a list of Abbreviations please.

Author Response

Referee 1

Many thanks for your considered and useful comments.

Point 1 and 2

The abstract is in line with the Journal’s requirements. No changes made.

Graphical abstract,  done (submitted to editorial office)

Points 2 and 4 This review is a brief update on a continually changing field, I do not agree that the suggested Table and/or glossary will add anything to the manuscript.

Point 5 Done – lines 403-4 in original manuscript and 415-421 in revised version.

Point 6 See original manuscript lines 64-107.

Point 7 Done – lines 411-415

Point 8 – not required by Journal

Reviewer 2 Report

Comments and Suggestions for Authors

This is a very nice review describing the use of 4 biologic therapies in treatment of severe refractory asthma. The paper is well-written, easy to follow and shows the most important information about the use of these compounds and clinical trials of such. The author provided a good introduction as well as brief but nice descriptions of the respective interleukins. There are only few issues that could be corrected in the paper to make it even better. First, consider to add some more information about severe refractory eosinophilic asthma. Second, add at the beginning of  each subparagraph the dosing and route of administration of the drugs. Third, add the missing citation in line 137 as well as a subparagraph in paragraph 5. Fourth, consider to add a few words why these drugs have been chosen as the main subject of the review. 

Author Response

Referee 2

Many thanks for your considered and useful comments.

First, consider to add some more information about severe refractory eosinophilic asthma.

Apologies, but this is the focus of the review, so it is not clear what additional information is needed. No changes made.

Second, add at the beginning of each subparagraph the dosing and route of administration of the drugs.

Done (This information is in the original manuscript, but has been added for dupilumab.)

Third, add the missing citation in line 137 as well as a subparagraph in paragraph 5.

Done

 Fourth, consider to add a few words why these drugs have been chosen as the main subject of the review. 

Please see lines 60-63 in original manuscript.